# Helminth Parasites among Rodents in the Middle East Countries: A Systematic Review and Meta-Analysis

**DOI:** 10.3390/ani10122342

**Published:** 2020-12-09

**Authors:** Md Mazharul Islam, Elmoubashar Farag, Mohammad Mahmudul Hassan, Devendra Bansal, Salah Al Awaidy, Abdinasir Abubakar, Hamad Al-Rumaihi, Zilungile Mkhize-Kwitshana

**Affiliations:** 1Department of Animal Resources, Ministry of Municipality and Environment, Doha, P.O. Box 35081, Qatar; 2School of Laboratory Medicine and Medical Sciences, College of Health Sciences, University of KwaZulu Natal, Durban 4000, South Africa; 3Ministry of Public Health, Doha, P.O. Box 42, Qatar; dbansal@moph.gov.qa (D.B.); halromaihi@moph.gov.qa (H.A.-R.); 4Faculty of Veterinary Medicine, Chattogram Veterinary and Animal Sciences University, Khulshi, Chattogram 4225, Bangladesh; miladhasan@yahoo.com; 5Ministry of Health, P.O. Box 393, PC 113 Muscat, Oman; salah.awaidy@gmail.com; 6Infectious Hazard Preparedness (IHP) Unit, WHO Health Emergencies Department (WHE), World Health Organization, Regional Office for the Eastern Mediterranean, Cairo 11371, Egypt; abubakara@who.int; 7School of Life Sciences, College of Agriculture, Engineering & Science, University of KwaZulu Natal, Durban 40000, South Africa; mkhizekwitshanaz@ukzn.ac.za; 8South African Medical Research Council, Cape Town 7505, South Africa

**Keywords:** rodent, helminth, cestode, trematode, nematode, Middle East, meta-analysis

## Abstract

**Simple Summary:**

The review was conducted to establish an overview of rodent helminths in the Middle East as well as their public health importance. Following a systematic search, 65 field research were identified, studied, and analyzed. The overall prevalence of cestodes, nematodes, and trematodes were 24.88%, 32.71%, and 10.17%, respectively. The review detected 21 species of cestodes, 56 nematodes, and 23 trematodes, from which 22 have zoonotic importance. *Capillaria hepatica*, *Hymenolepis diminuta*, *Hymenolepis nana*, and *Cysticercus fasciolaris* were the most frequent and widespread zoonotic helminths. The review identified that there is an information gap on rodent helminths at the humans-animal interface level in this region. Therefore, the public health importance of rodent-borne helminth parasites is not fully recognized. Countrywide detailed studies on rodent helminths, along with the impact on public health, should be conducted in this region.

**Abstract:**

Rodents can be a source of zoonotic helminths in the Middle East and also in other parts of the world. The current systematic review aimed to provide baseline data on rodent helminths to recognize the threats of helminth parasites on public health in the Middle East region. Following a systematic search on PubMed, Scopus, and Web of Science, a total of 65 research studies on rodent cestodes, nematodes, and trematodes, which were conducted in the countries of the Middle East, were analyzed. The study identified 44 rodent species from which *Mus musculus*, *Rattus norvegicus*, and *Rattus rattus* were most common (63%) and recognized as the primary rodent hosts for helminth infestation in this region. Cestodes were the most frequently reported (*n* = 50), followed by nematodes (49), and trematodes (14). The random effect meta-analysis showed that the pooled prevalence of cestode (57.66%, 95%CI: 34.63–80.70, *l^2^*% = 85.6, *p* < 0.001) was higher in Saudi Arabia, followed by nematode (56.24%, 95%CI: 11.40–101.1, *l^2^*% = 96.7, *p* < 0.001) in Turkey, and trematode (15.83%, 95%CI: 6.25–25.1, *l^2^*% = 98.5, *p* < 0.001) in Egypt. According to the overall prevalence estimates of individual studies, nematodes were higher (32.71%, 95%CI: 24.89–40.54, *l^2^*% = 98.6, *p* < 0.001) followed by cestodes (24.88%, 95%CI: 19.99–29.77, *l^2^*% = 94.9, *p* < 0.001) and trematodes (10.17%, 95%CI: 6.7–13.65, *l^2^*% = 98.3, *p* < 0.001) in the rodents of the Middle East countries. The review detected 22 species of helminths, which have zoonotic importance. The most frequent helminths were *Capillaria hepatica*, *Hymenolepis diminuta*, *Hymenolepis nana*, and *Cysticercus fasciolaris*. There was no report of rodent-helminths from Bahrain, Jordan, Lebanon, Oman, United Arab Emirates, and Yemen. Furthermore, there is an information gap on rodent helminths at the humans-animal interface level in Middle East countries. Through the One Health approach and countrywide detailed studies on rodent-related helminths along with their impact on public health, the rodent control program should be conducted in this region.

## 1. Introduction

Helminths are among the most diverse and geographically widespread groups of parasites that infect both humans and animals [1]. Although they are from different phyla or class (nematode, cestode, and trematode), the mode of transmission, infection, and pathogenesis, as well as host immune-responsiveness of these pathogens, follows a typical pattern. Approximately one-third of the world population is infected with one or more types of helminths. From amongst 300,000 species of helminths that typically infect vertebrates, 287 of them infect humans, from which 95% are either zoonoses or have evolved from animal parasites [2]. About 100 of the zoonotic helminths cause asymptomatic infection or mild symptoms in humans, while only a small percentage of them cause severe or even fatal infections [1]. In resource-poor countries, livestock is a source of food, production, income source, and deposit of wealth. Parasite infections in animals indirectly affect human health through financial hardship and malnutrition. Based on the burden of death, sickness, and treatment cost for both humans and animals for helminth infestation, the zoonotic parasites’ socioeconomic burden was presented as high or low socioeconomic impact [3].

Rodents are significant sources of parasitic zoonosis in humans, serving as reservoirs and vectors of at least 70 zoonotic diseases, of which 16 are helminth parasites [4]. Consumption of uncooked/improperly cooked food contaminated with the infective larvae, eggs, or metacercariae is the primary source of humans infestation with helminth parasites [5,6]. When pilfering humans food, rodents pass stool or urine that contaminates said food, leading to transmission of zoonotic helminths from rodents to humans [7].

The Middle East is an intercontinental region with a total population of over 411 million [8] in 17 sovereign countries, including Bahrain, Cyprus, Egypt, Iran, Iraq, Israel, Jordan, Kuwait, Lebanon, Oman, Palestine, Qatar, Kingdom of Saudi Arabia (KSA), Syria, Turkey, United Arab Emirates (UAE), and Yemen. The majority of people in this region live in poverty [9], with the highest percentage being specific to Yemen, Syria, Egypt, Palestine, and Iraq [10,11]. Cultural diversity, weak economic policy, poor governance, rapid population growth, low educational structure, gender discrimination, underdeveloped infrastructure, and war and conflict have turned the region into a hot spot for many emerging and re-emerging diseases, including rodent-borne parasitic infections [9,12,13]. In the past, rodent-borne infections have led to multiple instances of a fatal epidemic, in part due to a lack of relevant information available on the subject, which makes it difficult to maintain public health sustainability [14,15].

Helminth infestations are mostly neglected diseases [16]. Therefore, the complete picture of zoonotic helminths is not well known in the Middle East area. Despite several studies being done on helminths in this region, no systematic review or meta-analysis was performed on rodent helminths, including zoonotic importance in the Middle East region. Our objective is to summarize baseline information on rodent helminths in this region using evidence-based records of the helminths detected in rodents in the Middle Eastern countries. The review also identifies the rodent helminths with public health importance in this region.

## 2. Materials and Methods

We followed PRISMA (Preferred Reporting Items for Systematic Reviews and Meta-Analysis) guidelines [17] to conduct the systematic review using a four-step approach: database search, evaluating relevant articles, data extraction, and summarizing. One author conducted the data search. Two authors were involved in critical evaluation and data extraction from the selected articles, while one author managed the compilation of said data. Afterward, two authors arranged the data and conducted the meta-analysis (Figure 1, Appendix A).

### 2.1. Search Strategy

A literature search on rodent helminth parasites in the Middle East was performed on 17 June 2019 through PubMed, Web of Science, and Scopus (Figure 1). The search included all the original research articles containing field evidence of helminth parasites (trematode, nematode, cestode) among rodents in the Middle East countries. The search did not have any date range of publication. The keywords included (Rodent OR Rat OR Jird OR Gerbil OR Vole OR Mouse OR Hamster OR Porcupine OR Squirrel OR Jerboa) AND (Endoparasite OR Helminth OR Cestode OR Trematode OR Nematode) AND (17 Middle East country names individually). Screening on the search was conducted as [Title/Abstract] in PubMed, [TITLE-ABS-KEY] in Scopus, and [Topic] in Web of Science.

### 2.2. Search of Relevant Articles 

The data search results were processed using EndNote X9 (clarivate analytics, Philadelphia, PA, USA), which was also used to identify and exclude duplicate studies. Then we proceeded to peruse through the titles and abstracts to find the relevant articles. However, articles that were ambiguous regarding their relevance by their title and abstract were subjected to full-text analysis. Only documents published in English were considered for the review [18,19,20,21,22,23,24,25,26,27,28,29,30,31,32,33,34,35,36,37,38,39,40,41,42,43,44,45,46,47,48,49,50,51,52,53,54,55,56,57,58,59,60,61,62,63,64,65,66,67,68,69,70,71,72,73,74,75,76,77,78,79,80,81,82]. 

### 2.3. Data Extraction and Summarizing 

Evidence-based field reports give a clear picture of any pathogen’s availability, diversity, and dynamics in a locality [83,84]. We considered only the field reports containing rodent helminths for data abstraction. The extracted data included several variables such as country and location of sampling, season, year of sampling, rodent information (rodent species, sex, total rodent count, and the number of infected), helminth species and type, and possible associating factors of rodent infestation with helminth (Appendix A). The zoonotic rodent-borne helminths in this region were identified from the list of rodent helminths from this review with the support of published articles.

### 2.4. Data Analysis

The aggregated data was transcribed and stored in a Microsoft Excel spreadsheet, and then the data was forwarded to STATA/IC-13.0 (Stata Corp, 4905 Lakeway Drive, College Station, Texas 77845, USA) for statistical analysis. Crude prevalence estimation was performed by dividing the total number of helminth-positive rodents with the total number of rodents sampled and expressed as a percentage. The crude estimate of prevalence was used throughout, the 95% confidence interval (CI), and the *p*-value were calculated on different types of helminths among the countries. Study variations among the studies were evaluated using the Chi-square (χ2) test on Cochran’s *Q* statistics (with *p*-value) followed by *I*^2^ statistics to determine the study’s degree of heterogeneity. Standard Error (SE) was calculated using a standard formula for proportion calculation. A random-effect meta-analysis model was applied using the “mean” command specifying random due to the study’s high degree of heterogeneity (*I*^2^ > 75%). The output has been illustrated using a forest plot [85].

## 3. Results

### 3.1. Descriptive Analysis

The literature search returned 65 articles (Figure 1, Appendix A) published from 1969 to 2019. These articles were from 11 out of 17 Middle East countries, such as Cyprus, Egypt, Iran, Iraq, Israel, Kuwait, Palestine, Qatar, Saudi Arabia, Syria, and Turkey (Figure 2). No report on rodent helminths was available from Bahrain, Jordan, Lebanon, Oman, United Arab Emirates, and Yemen. Cestodes were the most frequently reported (50 articles) helminths in the Middle Eastern rodents, followed by nematodes (49), and trematodes (14). 

All 65 studies reported at least 9628 rodents (47% females and 53% males). A total of 44 rodent species from 6 families were listed (Appendix A). The analysis identified *Acomys dimidiatus*, *Jaculus jaculus*, *Meriones crassus*, *Mus musculus*, and *Rattus norvegicus*, *Rattus ratus* were widely distributed as these rodents were reported from where *Mus musculus* (*n* = 1251, 12.6%), *Rattus norvegicus* (*n* = 3325, 33.6%), and *Rattus rattus* (*n* = 1694, 17.1%) as the most common. Besides, three sub-species of *Rattus rattus*, such as *Rattus rattus alexandrines*, *Rattus rattus frugivorous*, and *Rattus rattus rattus*, are prevalent in the Middle East. Moreover, the review found some other rodents, which are important as zoonotic helminth carrier. These include *Acomys cahirinus*, *Acomys dimidiatus*, *Apodemus sylvaticus*, *Apodemus witherby*, *Arvicanthus niloticus*, *Calomyscus elburzensis*, *Cricetulus migratorius*, *Gerbillus cheesmani*, *Gerbillus gerbillus*, *Meriones libycus*, *Meriones persicus*, *Mesocricetus auratus*, *Microtus socialis*, *Microtus transcaspicus*, *Mus domesticus*, *Rhombomys opimus*, and *Tatera indica.* A total of 100 species of rodent helminths were identified. Based on the available data, the estimated pooled prevalence of the different types of parasites in rodents has been presented in Table 1. The random effect meta-analysis showed that the pooled prevalence of cestode ranged from 12.87% (95%CI: 5.17–20.57, *l*^2^% = 80.6, *p* < 0.001) in Turkey to 57.66% (95%CI: 34.63–80.70, *l*^2^% = 85.6, *p* < 0.001) in Saudi Arabia. The nematode prevalence was varying from 0.16% (95%CI: −0.15–0.47, *l*^2^% = 0.0) in Cyprus to 56.24% (95%CI: 11.40–101.1, *l*^2^% = 96.7, *p* < 0.001) in Turkey. Moreover, the prevalence of trematode ranged from 0.24% (95%CI: −0.11–0.59 *l*^2^% = 0.0, *p* < 0.001) in Iran to 15.83% (95%CI: 6.25–25.1, *l*^2^% = 98.5, *p* < 0.001) in Egypt. 

### 3.2. Rodent Cestodes in the Middle East Countries

Rodent cestodes information was available from all 11 Middle Eastern countries (Appendix A). A total of 21 rodent cestode species that belongs to 8 families have been reported in this review. Most of the cestodes were from Egypt and Iran (12 cestode species from each county). Out of 44 rodent species, *Mus musculus*, *Rattus norvegicus*, and *Rattus rattus* were frequently identified with the cestode infestation. Three species of cestodes have been frequently reported, viz: *Hymenolepis diminuta* (20 reports from 5 countries), *Hymenolepis nana* (30, 9), and *Cysticercus fasciolaris* (23, 4). Figure 3 shows the prevalence estimates from individual studies on cestodes in rodents of the Middle East countries, which ranged from 7.69 (95%CI: 2.22–13.17) to 68.57 (95%CI: 59.69–77.45) with an overall estimated prevalence 24.88 (95%CI: 19.99–29.77, *l*^2^% = 94.9, *p* < 0.001). 

### 3.3. Rodent Nematodes in the Middle East Countries

Rodent nematodes were studied in 8 countries in the Middle East, namely Cyprus, Egypt, Iran, Iraq, Israel, Kuwait, Palestine, and Turkey (Appendix A). Nematodes from 23 families represented the 56 nematode species in this region. Most of the rodent nematodes were reported from Egypt (*n* = 24) and Iran (*n* = 31) and the rodent species such as *Mus musculus*, *Rattus norvegicus*, *Rattus rattus*, *Meriones persicus*, *Acomys dimidiatus*, and *Tatera indica*. However, the nematodes, *Aspiculuris tetraptera*, *Capillaria hepatica*, *Syphacia obvelata*, *Streptopharagus kuntzi*, and *Trichuris muris* were most frequently reported and widely distributed. These nematodes were reported from three or more countries in the Middle East. Figure 4 shows the prevalence estimates from individual studies on nematodes in rodents of the Middle East countries, which ranged from 0.16 (95%CI: −0.15–0.47) to 79.41 (95%CI: 65.82–93.0) with an overall estimated prevalence of 32.71 (95%CI: 24.89–40.54, *l*^2^% = 98.6, *p* < 0.001).

### 3.4. Rodent Trematodes in the Middle East Countries

The reviewed studies reported rodent trematodes in Egypt, Iran, Israel, and Saudi Arabia (Appendix A). At least 23 trematode species from 11 families of trematodes were reported in the Middle Eastern rodents. Reports from Egypt (*n* = 21) were more descriptive of these trematodes. Moreover, *Fasciola* sp. was detected in Saudi Arabia, *Scaphiostomum* sp. in Israel, and *Notocotylus neyrai* and *Plagiorchis muris* were identified in Iran. The review found *Arvicanthus niloticus*, *Rattus norvegicus*, and *Rattus rattus* are three rodent species important for trematode infestation. Figure 5 shows the prevalence estimates from individual studies on trematode in rodents of the Middle East countries, which ranged from 0.20 (95%CI: −0.19–0.59) to 36.90 (95%CI: 26.59–47.22) with an overall estimated prevalence of 10.17 (95%CI: 6.7–13.65, *l*^2^% = 98.3, *p* < 0.001). 

### 3.5. Zoonotic Importance of the Rodent Helminths in the Middle East Countries

Out of the 100 species of rodent helminths detected in this review, 22 species have zoonotic importance; 7 cestodes, 6 nematodes, and 9 trematodes. The zoonotic helminths, their hosts, and possible human infection sources have been illustrated in Table 2. 

## 4. Discussion

This study reviewed the literature published in English on helminths-infested rodents in the Middle East region. The majority of the studies (47 of 65) were from Iran and Egypt, most likely due to their long history of rodent-borne zoonotic disease (like murine typhus, plague, tularemia) epidemics, which resulted in millions of death [4,91,92,93,94]. Thus, this topic became a central focus of public health research in these countries. The present review found three commensal rodent species: *Mus musculus*, *Rattus norvegicus*, and *Rattus rattus* to be more common and carrying most of the zoonotic helminths within this region. Previous literature described that these species occupy different habitats with higher population density than the other species and pose considerable risk to public health [4]. Although rodent cestodes were most frequently reported (*n* = 50) helminth in this review, the meta-analysis detected the overall rodent nematode prevalence was highest (32.7%) compared to cestodes (24.88%) and trematodes (10.17%) prevalence. Out of the 22 zoonotic helminths detected in this review, *Capillaria hepatica*, *H. diminuta*, *H. nana*, and *C. fasciolaris* have been found as widespread distribution. Furthermore, some non-zoonotic helminths such as *Aspiculuris tetraptera*, *Syphacia obvelata*, *Streptopharagus kuntzi*, and *Trichuris muris* were reported from three or more countries in this region.

Rodents have several beneficiary activities in ecology, such as soil aeration and water absorption ability, biotic recovery, and insect control [4,95]. In this regard, the presence of healthy rodents is essential for ecology [96]. Helminths infestation in rodents affects their own health and can subsequently alter the rodent-environment ecology to a considerable degree. Moreover, rodent helminths are important for humans, livestock, and pet animal health. Hymenolepiais is a major zoonotic rodent cestode [6]. Fascioliasis is hazardous for livestock health as well as for humans [6,97]. The definite host of *Taenia taeniaeformis* is the cat, where a stage of this cestode lifecycle (the cystic form, *Cysticercus fasciolaris*) is completed in rodents. An increase of *Cysticercus fasciolaris* in rodents can increase the health risk of cats [98]. Thus, rodent helminths have an impact on the ecology as well as humans and animal health.

Rodent-borne zoonotic helminths incur significant socioeconomic losses, although the zoonotic helminths’ socioeconomic burden can differ from species to species [3]. *Hymenolepis diminuta* and *Hymenolepis nana* are major zoonotic cestodes [3,99]. *Trichostrongylus* sp. and *Trichuris trichiura* are generally considered as major nematode threats [100]. The socioeconomic burden caused by *Angiostrongylus cantonensis*, *Gongylonema pulchrum*, *Trichinella* sp., and *Capillaria hepatica* are likely to be very low [3].

There is an information gap on rodent-borne zoonotic helminths in the Middle East countries. Some zoonotic cases of helminths infestation were reported in rodents by some countries in the Middle East, but none involved humans who might have been infected with the same helminths. Humans hymenolepiasis were reported in Bahrain [101], Cyprus [102], Jordan [103], Oman [104], Palestine [105], Qatar [106], and Yemen [107]. The *Hymenolepis nana* is a common zoonotic helminth transmitted from rodents to humans and the prevalence ranged from 0.15% to 12.2% in some Middle East countries with prevalence of specific countries such as Jordan (1.8%) [103], Oman (5.9%) [104], Palestine (1.0%) [105], Qatar (0.15%) [106], and Yemen (12.2%) [107]. Egg of *Hymenolepis diminuta* was detected from soil samples of school playgrounds of Jordan [108]. There is no report of rodent hymenolepiasis within these countries. *Echinococcus* spp. is a major helminth for human health, which was detected in rodents of Egypt, Iran, and Turkey. Human cases of alveolar hydatid cysts were reported from Iran, Kuwait, Saudi Arabia, and Turkey [109,110].

The rodent lungworm, *Angiostrongylus cantonensis*, causes eosinophilic meningomyelitis in humans, reported in Israel [111]. Gongylonema infection is reported in humans [112] and dromedaries [113] from Iran. *Trichinella* was a widespread parasite infecting humans and other mammals, although the former makes for a poor host for said organism [6]. There are reports of humans trichinellosis from Iran [114], Israel [115], Lebanon [116], and Turkey [117]. Human cases of trichostrongyliasis infestation were reported in Egypt [118], Iran [119], Israel [120], and Turkey [121]. Eggs of *Trichostrongylus* sp. were detected from soil samples of public places of Jordan [108] and Iraq [122]. Humans reports of *Trichuris trichiura* are available from Bahrain [123], Egypt [118], Israel [120], Jordan [103], Oman [104], Palestine [105], Qatar [124], Saudi Arabia [125], Turkey [126], and Yemen [107]. *Trichuris muris*, the rodent whipworm, does not have any zoonotic importance. Eggs of *Trichuris* were found in the soil of the public place of Iraq [122]. *Trichuris trichiura* is not a rodent specific nematode. The report of *Trichuris trichiura* in Iranian rodents [67] may be a case of accidental infestation.

*Schistosoma* and *Fasciola* are two major humans trematodes globally [3]. The high prevalence of *Fasciola* was recorded in Egypt, Iran, and Yemen [127]. Human cases of schistosomiasis were noted in Egypt [3], Iran [128], Israel [129], Jordan [103], Saudi Arabia [130], Turkey [131], and Yemen [3,132], whereas *Heterophyes heterophyes* were reported from Egypt and Saudi Arabia [133,134]. There are non-humans (fish, dogs, and cats) reports of *Pygidiopsis genata*, *Haplorchis pumilio*, *Haplorchis yokogawai*, and *Heterophyes heterophyes* from Egypt, Iran, Iraq, Israel, Palestine, Kuwait, Saudi Arabia, Turkey, UAE, and Yemen [135,136,137,138]. However, these rodent trematodes in the current review were mostly reported from Egypt and Iran.

Based on the meta-analysis, the overall prevalence of rodent trematodes was less than that of nematodes and cestodes in the Middle East, which had received more emphasis in other similar reports [110,127]. Efficient management of water resources are important factor for prevalence of trematode prevalence [139,140]. The presence of deserts means shortage of surface water in some of the countries of Arabian Peninsula such as Bahrain, Kuwait, Oman, Qatar, Saudi Arabia, and United Arab Emirates [141], which may be the cause of shortage of aquatic intermediate hosts of trematodes in these countries. Therefore, rodent trematodes are less reported in these countries. More research should be conducted to find rodent-borne trematodes in the countries of this region.

The reviewed articles in the current study described some of the factors that can influence the population of rodent-borne helminths within the Middle East, necessitating a need to develop a plan of action to control rodent helminths. The abundance of rodent-borne helminths depends on the host organism’s prevalence and its distribution [29,44]. An increase in the rodent population may increase the risk of humans getting infected by rodent parasites [142]. Rodents who inhabit animal farms have easy access to animal feed, and thus, they can be considered a potential vector and reservoir of animal and zoonotic diseases where animals serve as hosts [26]. *Hymenolepis diminuta* in the rodent are linked with some insects as intermediate hosts, such as *Xenopsylla astia*, which has a clear seasonal pattern. In Qatar, a research found that rodents are more infested with *Hymenolepis diminuta* in summer due to the *X. astia* abundance [24]. Several studies reported that the prevalence of rodent helminths is increased with rodent age [23,24,35,51]. Rodent helminths infestation can change with rodent host species [48]. The nematode, *Syphacia obvelata*, was reported to be most abundant in *Mus musculus* [64].

Rodent population control is a primary way to control rodent zoonotic diseases [143,144]. The other contributing factors, such as rodent species, seasons of the year, intermediate host, rodent control management in the residential areas, and animal farms, should also be considered on rodent related zoonoses control. As most of the rodent-related zoonotic helminths are linked to herbivores and carnivores [5,6,86,88], it is vital to manage dogs, cats, and livestock animals to avoid the spread of helminth infestation. Thus, One Health practice comes as a practical approach to control rodent-borne helminth prevalence [145]. “One Health is a collaborative, multisectoral, and transdisciplinary approach - working at the local, regional, national, and global levels—with the goal of achieving optimal health outcomes recognizing the interconnection between people, animals, plants, and their shared environment” [146]. One Health practice by linking veterinary, medical, ecology, entomology, parasitology, zoology fields, and local people are essential for rodent helminths prevention and control.

## 5. Conclusions

Rodent helminths in the Middle Eastern countries have been documented, which also highlighted rodent-borne zoonotic helminths. *Rattus norvegicus*, *Rattus rattus*, and *Mus musculus* were the most frequently reported rodents and infected with helminth parasites. Out of the 22 rodent-related zoonotic helminths, *Capillaria hepatica*, *H. diminuta*, *H. nana*, and *C. fasciolaris* were most frequent in this region. The current study illustrates that there is an information gap on the availability, diversity, and dynamics of rodent helminths and their interaction between humans and animals in the Middle East. Thus, the public health importance of rodent-borne helminth parasites is not fully recognized. However, rodent control should be the primary concentration by a One Health approach to control the spread of these helminths at the humans-animal-environmental interface in the countries of this region. We also suggest countrywide and detailed studies be conducted on rodent-borne helminths along with their impact on public health in this region.

## Figures and Tables

**Figure 1 animals-10-02342-f001:**
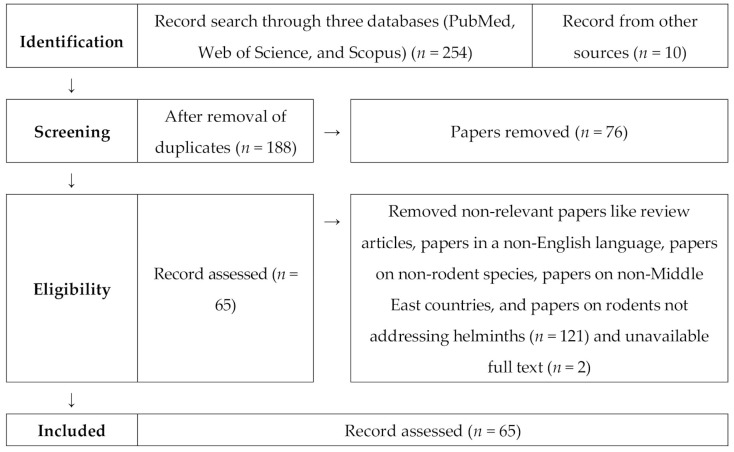
Systematic review PRISMA flow diagram describing the selection of published articles on rodent helminths in the Middle East and inclusion/exclusion process used in the study.

**Figure 2 animals-10-02342-f002:**
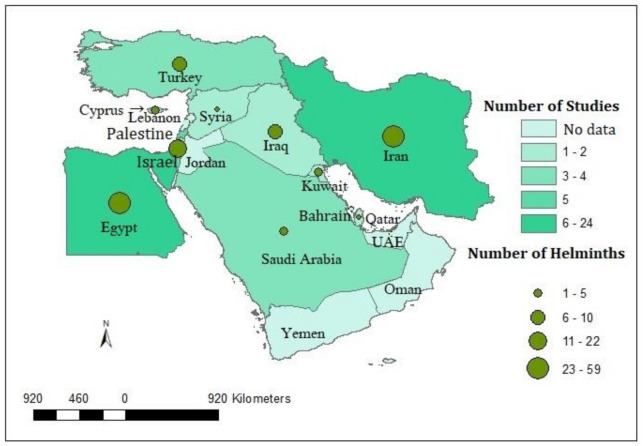
The map depicted the Middle East countries with the total number of studies and the number of helminths detected in rodents.

**Figure 3 animals-10-02342-f003:**
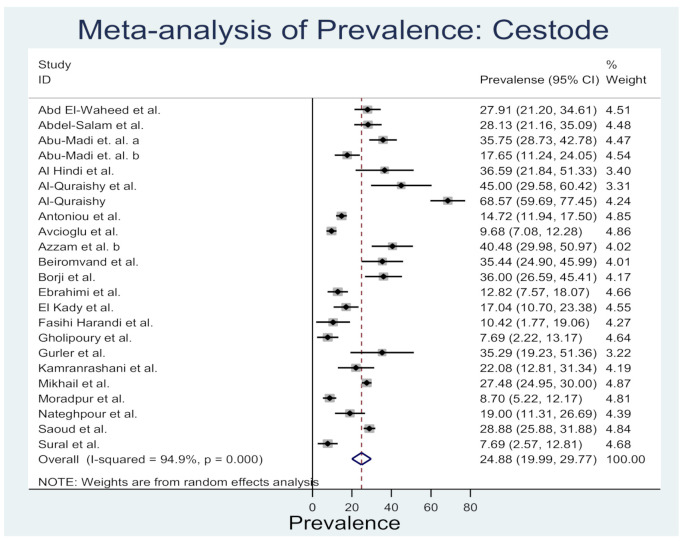
Forest plot of the prevalence estimates of cestode in rodents among the Middle East countries (the center dot representing point estimates whereas Gray Square representing the weight of each study to the meta-analysis).

**Figure 4 animals-10-02342-f004:**
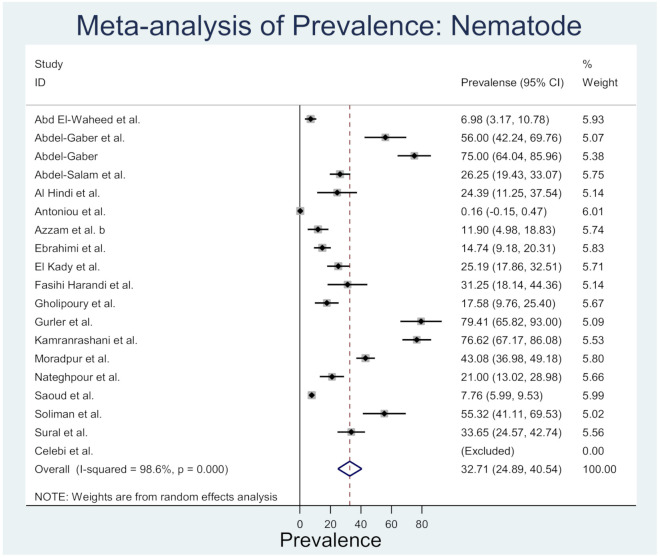
Forest plot of the prevalence estimates of nematode in rodents among the Middle East countries (the center dot representing point estimates whereas Gray Square representing the weight of each study to the meta-analysis).

**Figure 5 animals-10-02342-f005:**
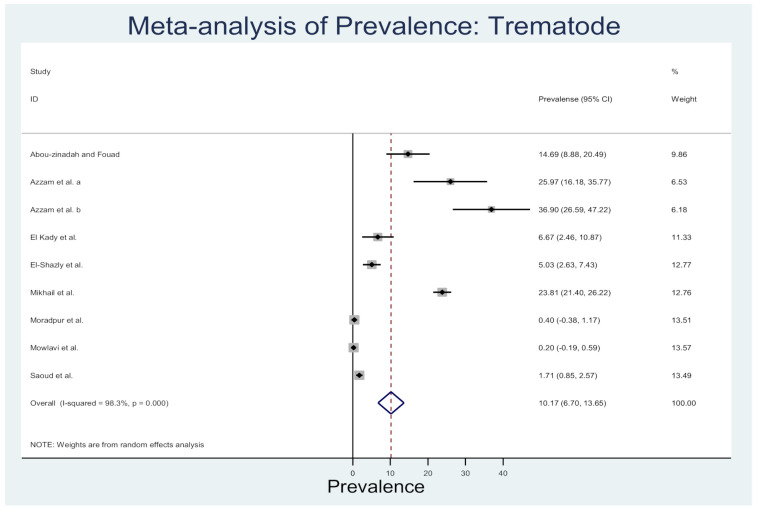
Forest plot of the prevalence estimates of trematode in rodents among the Middle East countries (the center dot representing point estimates, whereas Gray Square representing the weight of each study to the meta-analysis).

**Table 1 animals-10-02342-t001:** Estimated pooled prevalence of the rodent helminths in the Middle East countries.

Country	Parasite	Pooled Estimates (%)	95% CI	Heterogeneity Chi-Squared (χ2)	*l*^2^%	*p*-Value
Cyprus	Nematode	0.160	−0.15–0.47	0.00	0	-
	Cestode	14.72	11.94–17.50	0.00	0	-
Egypt	Nematode	31.81	19.83–43.78	259.62	97.3	<0.001
	Cestode	27.49	23.72–31.26	17.18	70.9	<0.001
	Trematode	15.827	6.56–25.1	344.74	98.5	<0.001
Iran	Cestode	18.21	11.59–24.83	56.08	87.5	<0.001
	Nematode	33.93	16.52–51.35	154.53	96.8	<0.001
	Trematode	0.24	−0.11–0.59	0.19	0	<0.001
Palestine	Nematode	24.39	11.25–37.54	0.00	0	-
	Cestode	36.59	21.84–51.32	0.00	0	-
Qatar	Cestode	26.64	8.89–44.39	13.94	92.8	<0.001
Saudi Arabia	Cestode	57.66	34.63–80.70	6.74	85.6	<0.001
	Trematode	14.685	8.88–20.49	0.00	0	-
Turkey	Nematode	56.24	11.40–101.1	30.10	96.7	<0.001
	Cestode	12.87	5.17–20.57	10.34	80.6	<0.001

CI: confidence interval; *I*^2^: inverse variance index; χ2: Cochran’s *Q* chi-square.

**Table 2 animals-10-02342-t002:** Rodent-borne zoonotic helminths in the Middle East countries.

Parasites	Host	Source of Human Infection	Reference
Rodent-borne zoonotic cestodes:			
*Raillieitina celebensis* and *R. demerariensis.*	DH: rodent; IH: ant and beetle	Ingestion of food contaminated with infected insects	[5,6]
*Hymenolepis diminuta* and *H. nana*	DH: rodent; IH: *H. diminuta*: flea and beetle. *H. nana* does not require IH.	Consumption contaminated food with rodent feces containing parasitic egg	[6,86,87]
*Mesocestoides* sp.	DH: dog and cat; 1st IH: ant and mite, 2nd IH: rodent, bird, amphibian, and reptile	Consumption of undercooked meat of amphibians and reptiles containing infective larva (tetrathyridium)	[5,6]
*Taenia taeniaeformis*	DH: cat; IH: rodent	There is a report that *Taenia taeniaformis* can infect humans	[88]
*Echinococcus multilocularis*	DH: dog, fox; IH: rat	Ingestion of embryonated eggs	[86]
Rodent-borne zoonotic nematodes:			
*Angiostrongylus cantonensis*	DH: rat and mollusk; IH: snail, prawn, crab, and frog	Ingestion of uncooked IH or vegetables contaminated with infected larvae	[6]
*Gongylonema pulchrum*	DH: ruminant, pig, wild boar, non-human primate, carnivore, and rodent; IH: beetles and cockroaches	Ingestion of IH or drinking of water contaminated with infective larvae	[5,6]
*Trichinella* spp.	Pig, wild boar, and rodent	Ingestion of uncooked muscle with encysted larvae	[6]
*Trichostrongylus* spp.	Herbivorous animal	Consumption of food and water contaminated with animal feces containing infective larvae	[6]
*Capillaria hepatica*	Rat, carnivore, and humans	Consumption of food contaminated with feces containing embryonated eggs	[5]
*Trichuris trichiura*	Humans	Consumption of food contaminated with feces containing *Trichuris* egg.	[5]
Rodent-borne zoonotic trematodes:			
*Echinochasmus* sp., *Echinoparyphium recurvatum*, and *Echoinostoma* sp.	DH: humans, rat, duck 1st IH: snail, 2nd IH: snail, amphibian, bivalve, fish	Ingestion of uncooked fish containing metacercariae	[2,89]
*Fasciola hepatica*	DH: herbivore; IH: snail	Ingestion of metacercariae contaminated vegetable	[6,34]
Haplorchis *pumilio*, *Pygidiopsis genata*, *Stictodora tridactyla*, *Prosthodendrium* spp., and *Plagiorchis muris*	DH: dog, cat, rat, duck, humans; 1st IH: snail, 2nd IH: fish	Eating uncooked fish harboring viable metacercariae	[88,90]
*Schistosoma mansoni*	DH: Vertebrate animal; IH: snail	Penetrate the DH skin	[6,86]

Note: DH: Definite host, IH: Intermediate host.

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
