# Peer review of "Helminth Parasites among Rodents in the Middle East Countries: A Systematic Review and Meta-Analysis"

_animals, 2020, doi:10.3390/ani10122342_

Round 1
Reviewer 1 Report
I find that this review is well written and its concepts are clearly
I did not find repetitions and I consider the results a good source to further clarify the role of rodents in the transmission of zoonoses.
Author Response
October 30, 2020
Dear Reviewer,
We are pleased to you for reviewing the manuscript entitled: Helminth parasites among rodents in the Middle East countries: A systematic review and Meta-analysis. Thank you for your appreciation on the manuscript.
We hope that the manuscript would be considered acceptable for publication in the ‘animals’.
Sincerely,
Md Mazharul Islam
Department of Animal Resources
Doha-Qatar
Reviewer 2 Report
Presented ms is a well prepared systematic review with a meta-analysis. The work has been conducted according to the PRISMA guidelines. The ms touches important topic, describing the prevalence of helminths in rodents in the Middle East.
I have some minor reservations to be solved before publishing.
Since English is not my first language, I usually refrain from advising text editing. In this case, I would kindly invite the authors to proofread the manuscript focusing particularly in the abstract, introduction and use special attention for the discussion.
There are some minor issues as follow:
Line 34 (abstract): please consider rephrasing "Rodents can be a source of zoonotic helminths in the Middle East.". Not only in the Middle-East, but also in other parts of the world.
Line 44 (abstract): Trematode (15.827 (95%CI: 6.25-25.1, l2%=98.5, p<0.001). Please add %sign after 15.827.
Line 54: "Rodent control programme" please consider changing capital letter.
Line 61: References needed.
Line 64: please rephrase or delete "Of these species of helminths"
Line 67: please rephrase "Sickness in animals..". Consider "Parasite infections in animals.."
Line 68: References needed.
Line 71-72: "as reservoirs and carriers". Please change carriers into vectors.
Line 74: please rephrase "rodent-related helminths"
Line 88: "Helminth infestation is mostly neglected diseases in the tropical region [16]" - hard to understand, rephrase, please.
Line 91: "In this regard" please remove. Keep language simple. do not use redundant sentences.
Line 180-181: Prevalence must be reported as a %. Please add missing % marks.
Line 192: "Trichuris (spp.) muris" - what species do you mean?
Line 221: Table 2. All abbreviations have to be explained. Please explain DH; IH; etc. It is easy for parasitologists but not for a general audience.
Line 242: "rodent-related helminths" please rephrase
Line 248: "Rodent-related zoonotic helminths" please rephrase
In my opinion, some crucial references are missing. Please consider including:
*Brown and Laco 2015 (https://www.ncbi.nlm.nih.gov/pmc/articles/PMC5570435/)
*Bordes et al 2015 (https://pubmed.ncbi.nlm.nih.gov/26176684/)
Overall, I recommend the article for publication only after revision of the English form and amendment of minor issues in a way that would better highlight obtained results.
Author Response
October 30, 2020
Dear Reviewer,
We are pleased to provide a revised manuscript entitled: Helminth parasites among rodents in the Middle East countries: A systematic review and Meta-analysis. Thank you for your overall enthusiasm for our study and the constructive comments, which have significantly improved the manuscript.
We corrected the manuscript, as you suggested, and detailed below based on your comments. We look forward to hearing from you about the review of our revised manuscript and hope that the manuscript would be considered acceptable for publication in the ‘animals’.
Sincerely,
Md Mazharul Islam
Department of Animal Resources
Doha-Qatar
Comments and Suggestions for Authors
Presented ms is a well prepared systematic review with a meta-analysis. The work has been conducted according to the PRISMA guidelines. The ms touches important topic, describing the prevalence of helminths in rodents in the Middle East.
Authors' reply: Thank you for reviewing the manuscript and escalating us on our writing.
I have some minor reservations to be solved before publishing.
Since English is not my first language, I usually refrain from advising text editing. In this case, I would kindly invite the authors to proofread the manuscript focusing particularly in the abstract, introduction and use special attention for the discussion.
Authors' reply: Thank you. A native speaker has reviewed the manuscript.
There are some minor issues as follow:
Line 34 (abstract): please consider rephrasing "Rodents can be a source of zoonotic helminths in the Middle East.". Not only in the Middle-East, but also in other parts of the world.
Authors' reply: You are right. The sentence has been re-written as "Rodents can be a source of zoonotic helminths in the Middle East and also in other parts of the world". (Current line 34-35)
Line 44 (abstract): Trematode (15.827 (95%CI: 6.25-25.1, l2%=98.5, p<0.001). Please add %sign after 15.827.
Authors' reply: Thank you. The sentence has been corrected accordingly as "Trematode (15.83%, 95%CI: 6.25-25.1, l2%=98.5, p<0.001)". (Current line 45)
Line 54: "Rodent control programme" please consider changing capital letter.
Authors' reply: Thank you. The sentence has been corrected accordingly as "rodent control program". (Current line 62)
Line 61: References needed.
Authors' reply: The documents of the two sentences "Although they are from different phyla or class (nematode, cestode, and trematode), mode of transmission, infection, and pathogenesis, as well as host immune-responsiveness of these pathogens, follows a typical pattern. Approximately one-third of the world population is infected with one or more types of helminths (Ref). From amongst 300,000 species of helminths that typically infect vertebrates, 287 of them infect humans, from which 95% are either zoonoses or have evolved from animal parasites [2]" were retrieved from a single source and the reference has been mentioned at the end of the second sentence. (Current line 68-73)
Line 64: please rephrase or delete "Of these species of helminths"
Authors' reply: "Of these species of helminths" has been deleted, and the whole sentence has been re-written as "About 100 of the zoonotic helminths cause asymptomatic infection or mild symptoms in humans, while only a small percentage of them cause severe or even fatal infections". (Current line 73)
Line 67: please rephrase "Sickness in animals..". Consider "Parasite infections in animals.."
Authors' reply: Thank you. The manuscript has been corrected accordingly. (Current line 76)
Line 68: References needed.
Authors' reply: The two sentences "In resource-poor countries, livestock is a source of food, production, income source, and deposit of wealth. Parasite infections in animals indirectly affect human health through financial hardship and malnutrition. Based on the burden of death, sickness, and treatment cost for both humans and animals for helminth infestation, the zoonotic parasites' socioeconomic burden was presented as high or low socioeconomic impact [3]." were retrieved from a single source and the reference has been mentioned at the end of the second sentence.
Line 71-72: "as reservoirs and carriers". Please change carriers into vectors.
Authors' reply: Thank you. The sentence has been rephrased as "Rodents are significant sources of parasitic zoonosis in humans, serving as reservoirs and vectors of at least 70 zoonotic diseases, of which 16 are helminth parasites". (Current line 80-81)
Line 74: please rephrase "rodent-related helminths"
Authors' reply: Thank you. The sentence has been rephrased as "Consumption of uncooked/improperly cooked food contaminated with the infective larvae, eggs, or metacercariae is the primary source of human infestation with helminth parasites". (Current line 82-83)
Line 88: "Helminth infestation is mostly neglected diseases in the tropical region [16]" - hard to understand, rephrase, please.
Authors' reply: Thank you. The sentence has been rephrased as "Helminth infestations are mostly neglected diseases". (Current line 97)
Line 91: "In this regard" please remove. Keep language simple. do not use redundant sentences.
Authors' reply: Thank you. The words "In this regard" has been removed. (Current line 133)
Line 180-181: Prevalence must be reported as a %. Please add missing % marks.
Authors’ reply: Thank you. The sentence has been corrected by adding % in the relevant places as “The random effect meta-analysis showed that the pooled prevalence of cestode ranged from 12.87% (95%CI: 5.17-20.57, l2%=80.6, p<0.001) in Turkey to 57.66% (95%CI: 34.63-80.70, l2%=85.6, p<0.001) in Saudi Arabia. The nematode prevalence was varying from 0.16% (95%CI: -0.15-0.47, l2%=0.0) in Cyprus to 56.24% (95%CI: 11.40-101.1, l2%=96.7, p<0.001) in Turkey. Moreover, the prevalence of trematode ranged from 0.24% (95%CI: -0.11-0.59 l2%=0.0, p<0.001) in Iran to 15.83% (95%CI: 6.25-25.1, l2%=98.5, p<0.001) in Egypt.” (Current line 208-212)
Line 192: "Trichuris (spp.) muris" - what species do you mean?
Authors' reply: Thank you. The species name has been corrected as "Trichuris muris". (Current line 238)
Line 221: Table 2. All abbreviations have to be explained. Please explain DH; IH; etc. It is easy for parasitologists but not for a general audience.
Authors' reply: Thank you. DH and IH have been explained as notes at the bottom of the table as "Note: DH: Definite host, IH: Intermediate host".
Line 242: "rodent-related helminths" please rephrase
Authors' reply: Thank you. The sentence has been rephrased as "rodent helminths are important for human, livestock, and pet animal health". (Current line 294)
Line 248: "Rodent-related zoonotic helminths" please rephrase
Authors' reply: Thank you. The sentence has been rephrased by replacing "Rodent-related zoonotic helminths" with "Rodent-borne zoonotic helminths" (Current line 306)
In my opinion, some crucial references are missing. Please consider including:
*Brown and Laco 2015 (https://www.ncbi.nlm.nih.gov/pmc/articles/PMC5570435/)
*Bordes et al 2015 (https://pubmed.ncbi.nlm.nih.gov/26176684/)
Authors' reply: Thank you. These two references have been applied in appropriate places; line 360 and line 350, respectively.
Overall, I recommend the article for publication only after revision of the English form and amendment of minor issues in a way that would better highlight obtained results.
Thank you for your suggestion.
Reviewer 3 Report
see attached

Author Response
October 30, 2020
Dear Reviewer,
We are pleased to provide a revised manuscript entitled: Helminth parasites among rodents in the Middle East countries: A systematic review and Meta-analysis. Thank you for your overall enthusiasm for our study and the constructive comments, which have significantly improved the manuscript.
We corrected the manuscript, as you suggested, and detailed below based on your comments. We look forward to hearing from you about the review of our revised manuscript and hope that the manuscript would be considered acceptable for publication in the ‘animals’.
Sincerely,
Md Mazharul Islam
Department of Animal Resources
Doha-Qatar
Specific responses to reviewers:
Comments and suggestion for authors
Rodents are the small mammalian animals with significant health concerns to human, which transmits not only the deadly infectious diseases such as murine typhus, plague and tularemia, but also many zoonotic parasitic diseases. Identification of helminth infections in rodent could provide information for their threat to human population in the region and evidence for prevention and control of their potential infections in human. Author systematically reviewed the helminth infections among rodents in the Middle East, identified the overall prevalence of cestodes, nematodes, and trematodes in rodents were 24.88%, 32.71%, and 10.17%, some of them have potential zoonotic importance to human infections in the region. The review is informative and provides evidence-based information for their important impact on public health in Middle East regions or other areas. I have some comments or concerns that should be addressed:
- The review provides overall profile of helminth infections in rodents, some of them can be transmitted to humans such as Capillaria hepatica, Hymenolepis diminuta, Hymenolepis nana, and Cysticercus fasciolaris. However, there is a gap between the prevalence of their infections in rodent and in human in this region. It will be much convincing if there is correlation data between the infections of these zoonotic helminths in rodents and the prevalence in human in the region. The epidemiological data of human infection of these rodent-born helminths should be included to determine their importance in public health in the region.
Authors reply: Thank you for your suggestion. This suggestion is out of the scope of the aim/hypothesis of the current manuscript. However, we added one sentences in the discussion part to explore the epidemiology of zoonotic parasites as suggested “The Hymenolepis nana is a common zoonotic helminth transmitted from rodents to humans and the prevalence ranged from 0.15% to 12.2% in some Middle East countries with prevalence of specific countries such as Jordan (1.8%) [103], Oman (5.9%) [104], Palestine (1.0%) [105], Qatar (0.15%)[106], and Yemen (12.2%) [107].” Line: 324 to 327.
- Page 10, Table 2, column of Source of human infection, several helminths just indicated the consumption of food contaminated with feces. The stage of the helminth contamination should be added, such as eggs, larvae of adult worms.
Authors' reply: Thank you. We updated the table 2 based on your suggestion
- Trichuris trichura should be human-specific nematode, only found in human or some non-human primate. It should not infect rodent
Authors' reply: Thank you. You are right; Trichuris trichiura is a human-specific nematode. However, there can be an accidental infection of Trichuris trichiura in non-human mammals. We conducted the review based on the published articles. There is one study that reported Trichuris trichiura in rodents of Iran (Nateghpour et al., 2014; PMID: 26114139). Another report mentioned Trichuris trichiura in dogs of Thailand (Areekul et al., 2010; DOI: 10.2478/abm-2010-0006. Of 17 Trichuris-positive dog feces samples examined by microscopy, 14 gave positive results by the PCR method in which four samples gave products for T. vulpis-specific primers (TT18SF and T18SR) while the remaining 10 samples yielded positive tests for T. trichiura specific primers (TT18SF and T18SR). However, in our manuscript, we mentioned as "Trichuris trichiura is not a rodent specific nematode. The report of Trichuris trichiura in Iranian rodents [67] may be a case of accidental infestation." Moreover, we corrected the host section of Trichuris trichiura in table 2. We kept only 'human' as host of Trichuris trichiura, and rodent and dog have been deleted.
- Ancylostoma ceylanicum is an important zoonotic nematode that infects both human and animals such as dog, cat and hamster. It should be involved in the review.
Authors' reply: Thank you. We worked only on the published documents of rodent helminths in the countries of the Middle East. As there is no report of Ancylostoma ceylanicum in the middle East rodents, we did not mention this nematode in the manuscript.
- It will be better if you could mention that some rodent nematode infections are the good laboratory models for human nematode infection, such as Trichuris muris mouse model for human trichuriasis, murine hookworm model Heligmosomoidespolygyrus bakeri and A. ceylaminum for human hookworm research, etc.
Authors' reply: Thank you. Our study objective was to (1) summarize baseline information on rodent helminths in the Middle Eastern countries based on evidence-based field reports and (2) identify the rodent helminths with public health importance in this region; therefore, we did not concentrate the laboratory related studies.